# A New Algorithm for Ill-Posed Problem of GNSS-Based Ionospheric Tomography

Debao Wen [1],*[ORCID], Kangyou Xie [1], Yinghao Tang [1], Dengkui Mei [2], Xi Chen [1] and Hanqing Chen [1]

1 School of Geography and Remote Sensing, Guangzhou University, Guangzhou 510006, China
2 School of Geodesy and Geomatics, Wuhan University, Wuhan 430072, China
* Correspondence: wdbwhigg@gzhu.edu.cn

**Abstract:** Ill-posedness of GNSS-based ionospheric tomography affects the stability and the accuracy of the inversion results. Truncated singular value decomposition (TSVD) is a common algorithm of ionospheric tomography reconstruction. However, the TSVD method usually has low inversion accuracy and reconstruction efficiency. To resolve the above problem, a truncated mapping singular value decomposition (TMSVD) algorithm is presented to improve the reconstructed accuracy and computational efficiency. To authenticate the effectiveness and the advantages of the TMSVD algorithm, a numerical test scheme is devised. Finally, ionospheric temporal–spatial variations of the selected reconstructed region are studied using the GNSS observations under different geomagnetic conditions. The reconstructed results of TMSVD can accurately reflect semiannual anomalies, diurnal variations, and geomagnetic storm effects. In contrast with the ionosonde data, it is found that the reconstructed profiles of the TMSVD method are more consistent with than those of the IRI 2016. The study suggests that TMSVD is an efficient algorithm for the tomographic reconstruction of ionospheric electron density (IED).

**Keywords:** ill-posed problem; ionospheric electron density; algorithm; computerized ionospheric tomography

## 1. Introduction

Ionosphere is an ionized component of the atmosphere over the earth. It is well known that the variation mechanism of the ionosphere is complex. The ionospheric delay phenomenon will occur when the electromagnetic signal penetrates the ionosphere, and the ionospheric delay error is an important error source in the fields of communication, satellite navigation and positioning, and radio science and space geodesy [1–3]. Therefore, it is necessary to grasp the temporal–spatial variation rules of the ionosphere. Fortunately, the constructions of global navigation satellite systems (GNSS) provide a promising means for the ionosphere sounding. Due to the real-time, continuous operation and global coverage characteristics of GNSS, GNSS has unique advantages over other ionospheric sounding techniques [4–6].

Vertical total electron content (VTEC) and ionospheric electron density (IED) are the frequently used ionospheric sounding parameters [7,8]. In general, VTEC can be obtained by using the thin-layer ionospheric model, which ignores the vertical structure of the ionosphere. Therefore, VTEC is usually used to describe the horizontal variations of the ionosphere [9–11]. To obtain high accuracy ionospheric delay correction, it is necessary to reconstruct a three-dimensional IED distribution.

Combining computerized ionospheric tomography (CIT) technique with simulated GNSS observations, Kunitsyn et al. have confirmed the possibilities of three-dimensional IED reconstruction [12]. However, ill-posedness of GNSS-based CIT is very prominent due to the nonuniformity and the sparsity of ground observation stations distribution [8,13–17]. To solve the above problem, some tomographic methods have been developed. In general,

the methods are usually divided into iterative and noniterative algorithms. Algebraic reconstruction technique (ART), multiplicative ART, and simultaneous iterative reconstruction technique (SIRT) are the typical representative of the iterative algorithms. Andreeva et al. introduced ART to study the ionospheric disturbance in Alaska during a geomagnetic storm at the end of October 2003 [18]. To overcome the deficiency of the conventional iterative algorithms, some constrained and improved iterative algorithms are proposed by some scholars [19–22]. In noniterative algorithms, singular value decomposition (SVD) and truncated SVD (TSVD) are usually adopted to reconstruct the IED distributions [23–27]. However, SVD and TSVD methods fail to obtain high accuracy solution since slant TEC (STEC) observations are contaminated by the discretized error and GNSS observation noise. Various approaches have been presented to overcome the problem that can be experienced with SVD and TSVD. In the approaches, modified SVD [28] and truncated generalized SVD [29] are the two typical algorithms. Although these approaches have their advantages over SVD and TSVD, they suffer from the difficulty of the regularization operator selection or the heavy computational effort. To overcome the disadvantage of the above-described algorithms, the truncated mapping SVD (TMSVD) method is proposed to obtain high-accuracy inversion results. The proposed TMSVD amalgamates information about the properties of the anticipated solution into the solutions process through an initial mapping of the IED tomographic reconstruction. The reconstructed accuracy and efficiency can be improved using the TMSVD. The numerical simulation results verify that the TMSVD is feasible to improve the reconstructed accuracy and computational efficiency. Finally, the TMSVD methods are successfully applied to reconstruct the IED images based on the actual GNSS observations. The error statistics and the comparisons of the vertical profiles further verify the advantages of the TMSVD.

## 2. Materials and Methods

### 2.1. Tomographic Theory

As is well known, GNSS-based CIT uses the input STEC to reconstruct the IED distribution [30]. The relation between STEC and IED can be represented using the following equation:

$$y_i = \int_p Ne(l) dl \tag{1}$$

where $Ne(l)$ is the reconstructed IED distribution. $y_i$ represents the STEC along ith ray path $p_i$. The actual IED $Ne$ can be approximated using a basis function $f_{ij}(l)$. Then $Ne$ is written as:

$$Ne(l) = \sum_{j=1}^{n} x_j f_{ij}(l) \tag{2}$$

where $x_j$ is the IED within the jth voxel. Substituting Equation (2) into Equation (1), Equation (1) is reformulated as:

$$y_i = \int_p \sum_{j=1}^{n} x_j f_{ij}(l) \, dl = \sum_{j=1}^{n} x_j \int_p f_{ij}(l) \, dl \tag{3}$$

The path integral of $f_{ij}(l)$ represents the length of the *i*th ray path traversing the *j*th voxel, which will be defined as $A_{ij}$.

$$A_{ij} = \int_p f_{ij}(l) \, dl \tag{4}$$

An indicator function is selected as the basis function, which is represented as:

$$f_{ij}(l) = \begin{cases} 1 & j\text{th voxel intersected by } p_i \\ 0 & \text{otherwise} \end{cases} \tag{5}$$

Then the shorter path integrals become

$$\sum_{j=1}^{n} A_{ij} x_j = y_i \qquad i = 1, 2, \cdots, m \tag{6}$$

Considering the discretized error and GNSS observation noise, the matrix expression of Equation (6) is as follows:

$$A_{m \times n} x_{n \times 1} + e_{m \times 1} = y_{m \times 1} \tag{7}$$

where $n$ is the number of the discretized voxels in the reconstructed area, $m$ is the number of the input STEC $y$ is a column vector of the $m$ known STEC values, $A$ is the coefficient matrix, and $x$ is the vector consisting of all the unknown IED in all the voxels [1].

*2.2. Tomographic Method*

To overcome the encountered difficulties of TSVD, the orthogonal mappings of the coefficient matrix $A$ and the input STEC vector $y$ are first performed, and then the TSVD is used to determine the approximation of the mapping problem. The above procedure is named TMSVD. The mapping divides the subspace of the inversion results into two sections. One section is obtained from the user; the other is computed using TSVD. The use-supplied section improved the reconstructed accuracy by incorporating prior information of the ionosphere, which is obtained from IRI 2016 model.

A subspace $\omega$ of $S_{n \times l}$ is first chosen, and the columns of the matrix W constitute an orthogonal basis of $\omega$. The following formula can be obtained by introducing QR decomposition.

$$AW = QR \tag{8}$$

where $W \in S_{n \times l}$, $Q \in S_{m \times l}$, $R \in S_{l \times l}$. In this case, $Q$ has orthogonal columns and $R$ is upper triangular. The selection of the subspace $\omega$ is to ensure that $AW$ is not rank deficiency. Therefore, the matrix $R$ is zero strangeness. Innovating the following orthogonal mapping factors:

$$\Psi_W = WW^T, \quad \Psi'_W = I - \Psi_W, \quad \Psi_Q = QQ^T, \quad \Psi'_Q = I - \Psi_Q$$

Then the solution $x$ of Equation (3) can be divided into two sections, which can be represented as:

$$\begin{cases} x = x' + x'' \\ x' = \Psi_W x \\ x'' = \Psi'_W x \end{cases} \tag{9}$$

The decomposition of Equation (3) can be written as:

$$\Psi_Q A x' + \Psi_Q A x'' = \Psi_Q y \tag{10}$$

In general, $\Psi'_Q A \Psi_W = 0$, so the following expression can be obtained:

$$\Psi'_Q A x'' = \Psi'_Q y \tag{11}$$

TSVD is innovated to fulfill the mapping of Equation (11). Since $\Psi'_Q A \Psi'_W = \Psi'_Q A$, the TSVD of $\Psi'_Q A$ can be performed. The approximate $x''_k$ of Equation (11) is obtained. The $k$ value is determined using deviation principle. Then the approximate solution $x'_k$ of Equation (10) is computed. The final expression can be represented as:

$$R z'_k = Q^T (y - A x''_k) \tag{12}$$

The solution $z'_k$ can be obtained, and then the solution $x'_k$ is inverted using the following equation:

$$x'_k = Wz'_k \tag{13}$$

The final solution of IED can be expressed as:

$$x_k = x'_k + x''_k \tag{14}$$

For TMSVD, the regularization is only carried out in the subspace $\omega$. Therefore, the subspace is selected in order that ill-posedness of the matrix $R = Q^T AW$ can be circumvented. For the ill-posed problem of discrete tomographic inversion, this condition is easily satisfied when $\omega$ represents smooth functions.

Let $x''_k$ represents the approximate solution of Equation (11). The linear system of Equation (12) is exactly solved for $z'_k$. The solutions of $x'_k$ and $x_k$ are solved using Equations (13) and (14), respectively. Then:

$$\|y - Ax_k\| = \|\Psi'y - \Psi'Ax''_k\| = \|y - Ax''_k\| \tag{15}$$

The solution of high accuracy can be determined since Equation (12) is not ill-posed.

## 3. Results

### 3.1. Numerical Simulation

To confirm the performance of TMSVD, a numerical scheme is first devised. Since TMSVD is the improvement of the common TSVD, the TSVD is introduced to compare with the new algorithm. The simulated process is as follows.

A numerical scheme is devised to illustrate the advantages of the TMSVD in comparison with the TSVD. In the numerical simulation, the true values of IED distribution are generated from IRI 2016 model. The selected time period is 05:30–06:00 UT, 30 October 2020. The latitudinal range is 30°–36°N, and the longitudinal range is 116°–122°E. In vertical direction, the altitudinal range is 100–700 km in steps of 10 km. The discretized intervals are 0.5° in latitude and longitude. Thus, the reconstructed region is divided into 4320 voxels. For the test of TMSVD, the space coordinates of the GNSS observation stations and the observed GNSS satellites are used to construct the matrix $A$.

To evaluate the advantage of the TMSVD to the common TSVD, the mean absolute error (MAE) and the root mean square error (RMSE) of the two algorithms can be calculated using Equations (16) and (17), respectively [8].

$$\text{MAE} = \sum_{j=1}^{n} \left| x_j^{true} - x_j^{tomo} \right| / n \tag{16}$$

$$\text{RMSE} = \sqrt{ \sum_{j=1}^{n} \left( x_j^{true} - x_j^{tomo} \right)^2 / n } \tag{17}$$

where $x_j^{true}$ is the simulated IED true value of the $j$th voxel, and $x_j^{tomo}$ represents the tomographic solution of the two algorithms.

Using the devised scheme, the two algorithms are used to reconstruct the IED distribution of the selected geographic region. Figure 1 illustrates the comparisons between the tomographic solutions of the above methods and the simulated IED true values. The comparisons confirm that the reconstructed IED distributions of TMSVD has better agreement with IED true values than those of TSVD.

The reconstructed error between the inversion results of the two algorithms and the simulated true values is computed. Figure 2 shows the error statistical diagram of the two algorithms. Figure 2a illustrates that the maximum error absolute value is $3.87 \times 10^{10}$ el/m³, and the error absolute values of about 85% voxels are less than $2 \times 10^{10}$ el/m³. Figure 2b shows that the maximum error absolute value of the error is $1.16 \times 10^{11}$ el/m³, and the

error absolute values of only 27% voxels are less than $2 \times 10^{10}$ el/m$^3$. The error statistics validate the accuracy of TMSVD as being higher than that of the TSVD. According to Equations (16) and (17), the MAE and RMSE of two algorithms is obtained. The MAE of TMSVD is $1.01 \times 10^{10}$ el/m$^3$, and the RMSE is $1.54 \times 10^8$ el/m$^3$. However, the MAE of TSVD is $4.45 \times 10^{10}$ el/m$^3$, and the RMSE of TSVD is $6.8 \times 10^8$ el/m$^3$. The above facts validate that the TMSVD is superior to the common TSVD in performing the tomographic reconstruction of IED distributions.

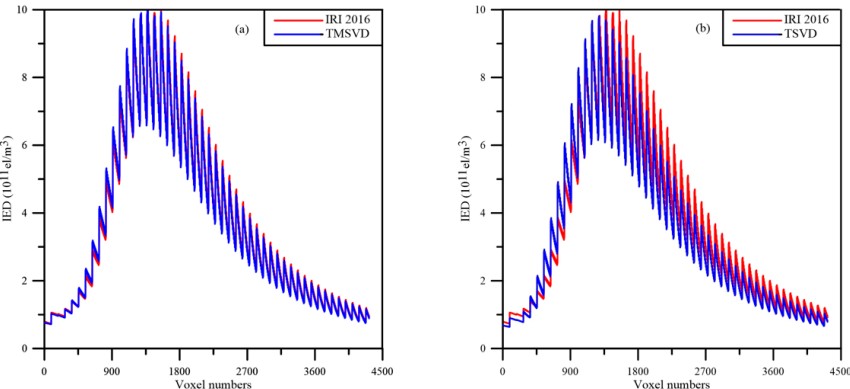

**Figure 1.** Comparisons of the tomographic results of two algorithms with the simulated IED true values. (**a**) TMSVD; (**b**) TSVD.

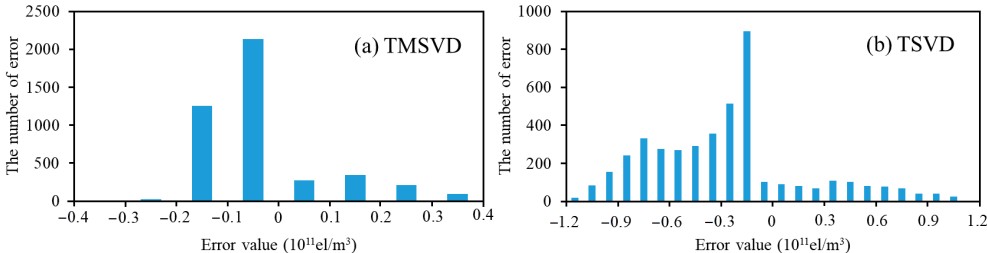

**Figure 2.** Error statistics of the TMSVD and the TSVD. (**a**) TMSVD; (**b**) TSVD.

*3.2. Test of TMSVD Based on GNSS Data*

To further test the performance of the TMSVD method, the GNSS observations is applied to reconstruct three dimensional IED distribution of the Hunan province. The sample interval of GNSS data is 30 s. The geographic positions of the selected GNSS ground stations are shown in Figure 3. To avoid the boundary distortion effect of the reconstructed imaging, the latitudinal and the longitudinal ranges are enlarged. For the selected reconstructed area, the latitude ranges from 24°N to 31°N with the step of 0.5°, the longitude ranges from 108°E to 115°E with the step of 1°, and the altitude ranges from 100 km to 1000 km with the spatial resolution of 50 km. Considering the small variation magnitude of IED between 650 km and 1000 km, the altitude ranges from 100 to 650 km when the IED distributions are reconstructed. In this work, the temporal resolution is 30 min.

Using the proposed TMSVD method, we investigate the IED variation rules under the conditions of geomagnetic quiet and disturbance. Figure 4 illustrates three dimensional IED distributions during geomagnetic quiet days on 4 January, 17 March, 17 June, and 18 September 2022. Figure 4 shows that the peak height of the ionosphere is 250 km on 4 January 2022. In the other three days, the peak heights appear at the altitude of 350 km. The reconstructed images capture the IED varied trends in the altitudinal direction.

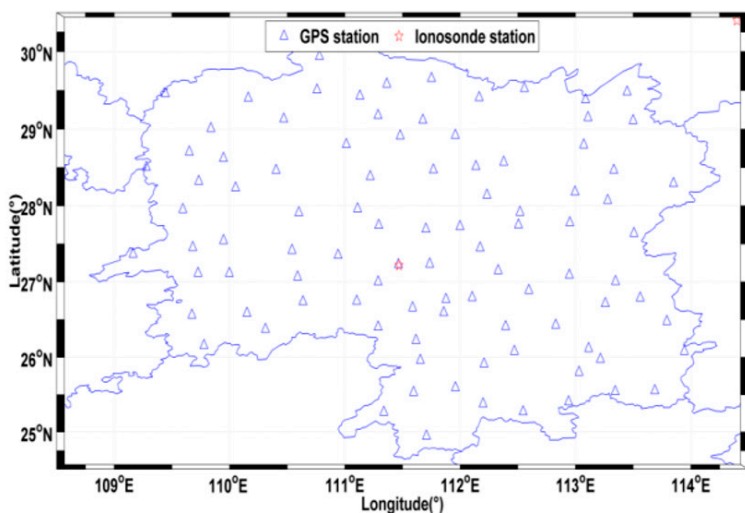

**Figure 3.** Locations of the selected GNSS and ionosonde stations.

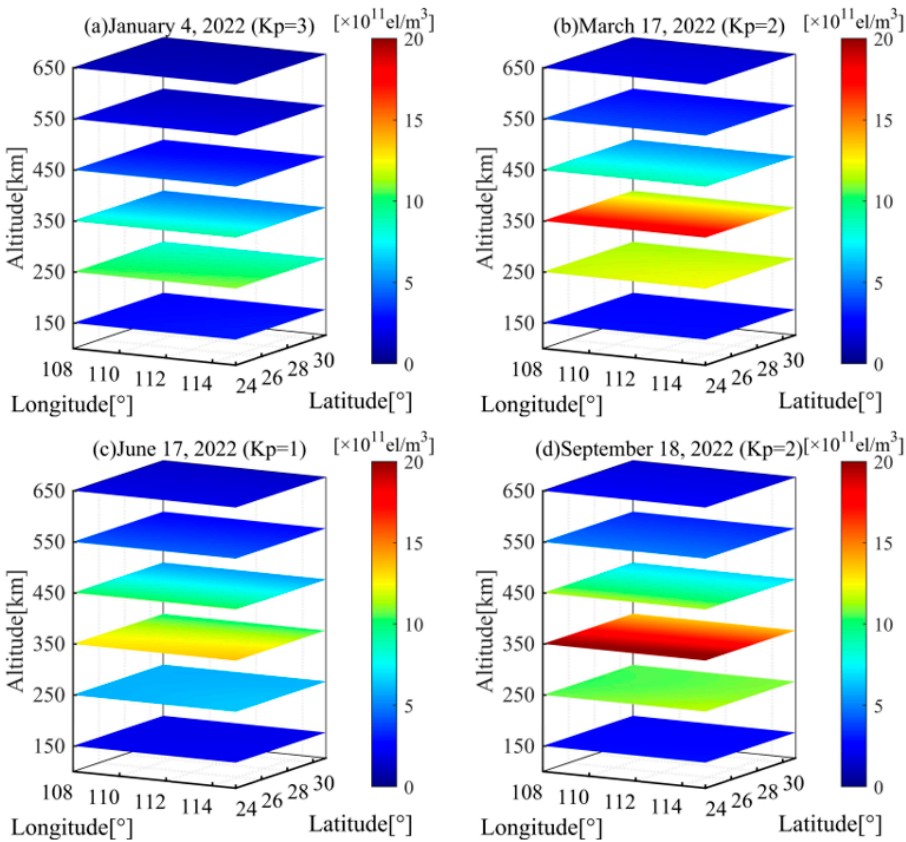

**Figure 4.** The horizontal section of the reconstructed IED distributions using TMSVD at the altitude of 150–650 km. (**a**) 5:00 UT on 4 January 2022; (**b**) 5:00 UT on 17 March 2022; (**c**) 5:00 UT on 17 June 2022; (**d**) 5:00 UT on 18 September 2022.

Figure 5 illustrates the vertical section at the latitude of 24°–31°N during the same time periods as Figure 4. It shows that the IED values in north Hunan are greater than those in south Hunan. In the meanwhile, the seasonal variations are unlocked. Comparing the reconstructed images in spring and fall with those in winter and summer in Figures 4 and 5, the IED values on 17 March and 18 September are greater than those on 4 January and 17 June 2022. The reconstructed results reflect the semiannual anomaly of three-dimensional ionospheric variations.

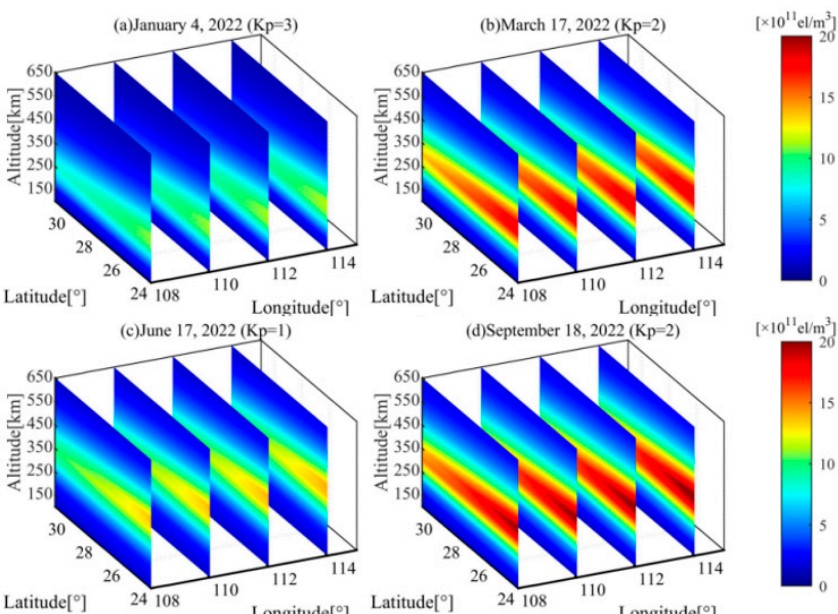

**Figure 5.** The three-dimensional IED distributions reconstructed by TMSVD at 05:00 UT. (**a**) 4 January 2022; (**b**) 17 March 2022; (**c**) 17 June 2022; (**d**) 18 September 2022.

Figure 6 reflects the diurnal variations of the ionosphere at the cross section of 350 km on 25 August 2018. According to the longitudinal distribution of the IED over the Hunan province, an apparent difference is shown. The IED values in the east is greater than those in the west during 01:00 UT–05:00 UT, and then the IED values in the west are greater than those in the east. However, during 21:00 UT–23:00 UT, the longitudinal distributions of the IED return to the state of the time periods of 01:00 UT–05:00 UT. In the latitudinal direction, the IED values in the north are greater than those in the south between 13:00 UT and 21:00 UT. Nevertheless, the IED values in the low-latitude region are greater than those in the high-latitude region in other time periods. As time goes on, the IED values gradually increase, and the IED value reaches its maximum at 07:00 UT. Subsequently, the electron density distributions start to decrease; the minimum IED value occurs at 21:00 UT (5:00 local time). The diurnal variations rules are identical to the Earth's rotation phenomenon. This also coincides with the alternation of day and night.

According to the statistics analysis, 4.15 Gb computer memory and 1400 CPU seconds are usually required when the TSVD is applied to reconstruct the diurnal variation in the IED distributions, whereas the TMSVD required 2.36 Gb computer memory and 650 CPU seconds. The statistics validate that the reconstructed efficiency of the TMSVD is higher than that of the TSVD.

An ionospheric storm occurred on 26 August 2018. The Kp index reached 7+ at 8:00 UT. To verify the ability of the TMSVD method to capture ionospheric storm, the above storm is selected as the test case. Figure 7d,e shows the IED distributions obtained from the TMSVD and the IRI 2016 model at 8:00 UT on 26 August 2018, respectively. Figure 7f reveals the difference between the results of the TMSVD and those of IRI 2016 model. Figure 7f manifests that the IED inversion results of the TMSVD are greater than those of the IRI 2016 model. Considering the quiet ionospheric activity on 25 August 2018, the three-dimensional IED distributions of the day are introduced to compare with those of the selected storm day. Figure 7a,b represents the reconstructed images of the TMSVD and IRI 2016 model, respectively. Figure 7c represents the difference between Figure 7a,b. Figure 7c shows that the reconstructed IED distributions become lower than the simulated IED values by the IRI 2016 model. Comparing Figure 7d with Figure 7a, the IED values of the storm day evidently increase. It exhibits the positive storm phase effect. Figure 7f shows that the positive storm phase covers the selected latitude range. However, the results obtained from the IRI 2016 model displays no visible difference on 25–26 August 2018.

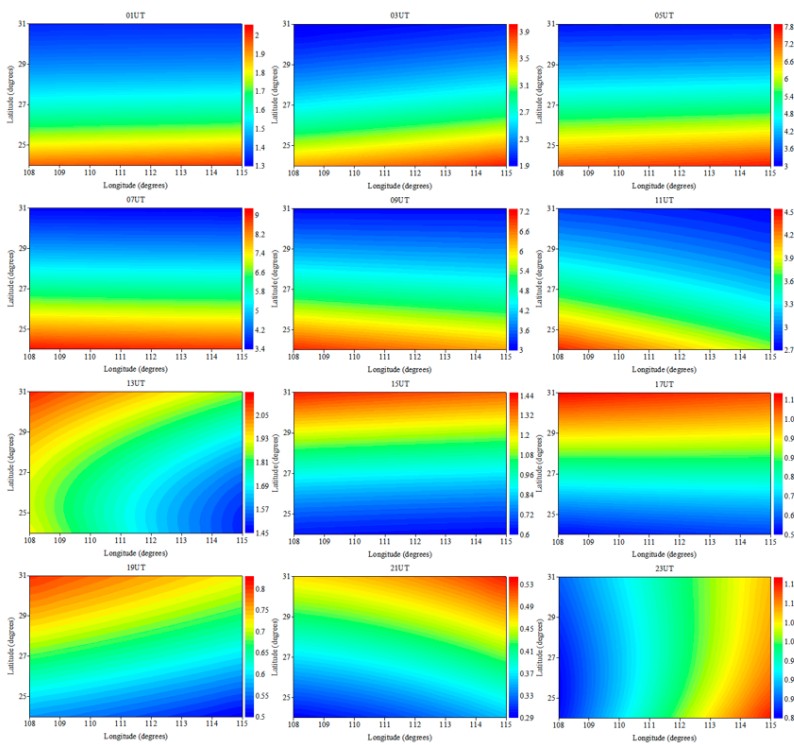

**Figure 6.** The IED diurnal variation at the altitude of 350 km on 25 August 2018. The unit of IED is $10^{11}$ el/m$^3$.

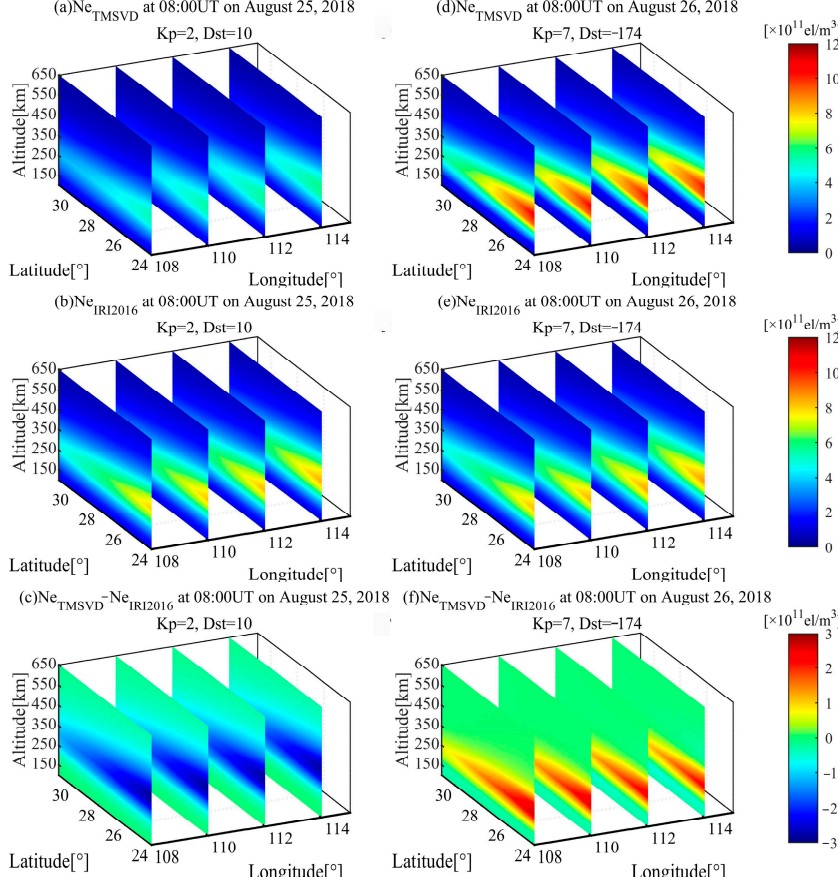

**Figure 7.** The three-dimensional IED distributions reconstructed using TMSVD and the IRI 2016 model at 8:00 on 25–26 August 2018.

As illustrates in Figure 8, the comparisons of the vertical IED profiles are made at the time mentioned in Figure 7. In the comparisons, the recorded data of ionosonde located at Shaoyang is introduced. In general, ionosonde diagnoses only the lower part of the ionosphere. In this work, the type of profile above the F2 layer peak is derived from the Chapman model. The comparisons show that the vertical profiles reconstructed by TMSVD are identical to those obtained from ionosonde in Shaoyang. The fact validates that the TMSVD is effective and superior to the IED reconstruction under different geomagnetic conditions.

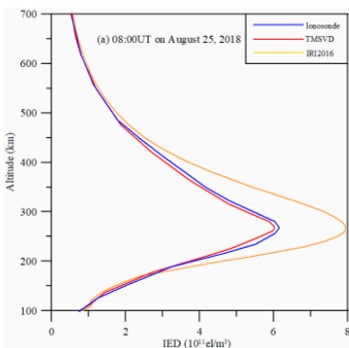 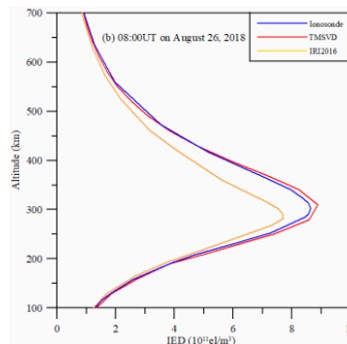

**Figure 8.** The comparisons of the IED profiles reconstructed by the TMSVD with those obtained from the IRI 2016 and ionosonde in Shaoyang.

## 4. Conclusions

This work develops a new algorithm to reconstruct the ionospheric structure by using the GNSS observation in the Hunan province. Two experimental schemes are devised to validate the advantages of the TMSVD method. The experiments confirm that the inversion accuracy and efficiency is obviously improved by the TMSVD method. Seasonal characteristics were first studied on 4 January, 17 March, 17 June 2022, and 18 September 2022. The reconstructed results of the TMSVD can accurately capture the semiannual anomalies and the diurnal variations. Compared to the corresponding values obtained from the IRI 2016 model, the reconstructed IED values of the TMSVD have an obvious improvement. Finally, a strong ionospheric storm is selected as a test case. The test results show that the TMSVD method can effectively capture the ionospheric structure during an ionospheric storm. However, the IRI 2016 model cannot reflect the IED variations under the condition of ionospheric disturbance.

Although the TMSVD can effectively reconstruct the three-dimensional IED distributions under different geomagnetic activities, the algorithm has difficulty in reconstructing the polar cap absorption event. In future, the TMSVD will be extended to reconstruct the solar flare and equatorial anomaly. In addition, the study of ionospheric activities caused by geohazards can be carried out.

**Author Contributions:** Methodology, D.W.; software, Y.T. and K.X.; validation, X.C. and D.M.; formal analysis, D.W.; writing—original draft preparation, D.W.; writing—review and editing, H.C.; funding acquisition, D.W. and H.C. All authors have read and agreed to the published version of the manuscript.

**Funding:** This research was funded by the National Nature Science Foundation of China, grant number 42070430, the Natural Science Foundation of Guangdong Province of China, grant number 2022A1515011039.

**Data Availability Statement:** The readers can obtain the data from the corresponding author.

**Acknowledgments:** All authors would like to thank the editors and anonymous reviewers for their valuable comments and suggestions which improved the quality of the manuscript.

**Conflicts of Interest:** The authors declare no conflict of interest.

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
