# Peer review of "A New Algorithm for Ill-Posed Problem of GNSS-Based Ionospheric Tomography"

_remotesensing, doi:10.3390/rs15071930_

Round 1
Reviewer 1 Report
Review on the article “A New Algorithm for Ill-posed Problem of GNSS-based Ionospheric Tomography”.
Author – Debao Wen et al.
Presented for publication in Remote Sensing (Atmosphere)
General comments:
The presented work is undoubtedly interesting as a methodological development in constructing accurate and correct solution to a poorly defined (Ill-posed) GNSS tomography problem. A new approach to solving the matrix equation for reconstructing the three-dimensional electron density distribution in TEC measurements on a certain plane, the earth's surface, is proposed. The stability and correctness of the solution is studied for the TEC data synthesized in the IRI-2016 ionosphere model and the results of applying the proposed method to the actual results of GNSS measurements in calm and disturbed ionospheric conditions are presented. A positive point in the work is the comparison with the data of practically direct measurements of the IED by an ionosonde of vertical radio sounding of the ionosphere in the positive phase of ionospheric storm at low latitudes. The paper is recommended for publication with taking into account remarks.
Specific Comments:
Title – The title reflects a sense of the article.
Abstract – The abstract generally reflects the problem description, aim and results of the article.
Introduction – Introduction describes the general problem of constructing a stable and correct solution in a poorly defined GNSS (TEC) tomography problem, proposed approach and the general result obtained in this paper.
Section 2: Method
A description of the proposed method is given.
Remarks. The essence of the method is not very clear. An addition is required for clarification in the formation of the Amn matrix. Apparently, it corresponds to the piecewise constant approximation of the IED. Also, an explanation is needed for the a priori initial approximation and the orthogonal basis. If a model is used, it must be specified what is the IED model.
Section 3: Numerical Experiment
Again, it is necessary to specify the initial approximation for the solution in a numerical experiment.
Section 4: Test of Tmsvd Based on GNSS Measured Data
VI ionosonde diegnoses only the lower part of the ionosphere and it is necessary to specify the type of profile above the F2 layer peak.
In Fig.3 it is necessary to correct Dst = 174.
Section 5: Conclusion
I have not any comments.
References
I have not any comments.
Reviewer 2 Report
Overall, the manuscript is very well written, very interesting, and reports a new algorithm, called Truncated Mapping Singular Value Decomposition (TMSVD, which is able to perform better (compered with other algorithms) and therefore to improve both reconstruction accuracy and computational efficiency under various geomagnetic conditions. The manuscript is certainly recommended for publication in its present form.
In the proofreading process, the authors should correct:
L37: “IED” needs to be defined here in the text
L54: “TSVD” needs to be defined here in the text
L58: “SVD” needs to be defined here in the text
L62: “TMSVD” needs to be defined here in the text
L145: ”Tmsvd” should read “TMSVD”
L154-155: “…the longitude ranges from 24°N to 31°N with the step of 0.5°, the latitude ranges from 108°E to 115°E with…” should read “…the latitude ranges from 24°N to 31°N with the step of 0.5°, the longitude ranges from 108°E to 115°E with…”
The authors are mixing longitudes with latitudes here.
